# *Prunus mume* Extract Inhibits SARS-CoV-2 and Influenza Virus Infection In Vitro by Directly Targeting Viral Particles

**DOI:** 10.3390/ijms26178487

**Published:** 2025-09-01

**Authors:** Mizuki Tokusanai, Koichiro Tateishi, Kanako Hirata, Nahoko Fukunishi, Yusuke Suzuki, Ryohei Kono, Sorama Natsumi, Chikara Kato, Susumu Takekoshi, Yoshiharu Okuno, Hirotoshi Utsunomiya, Norio Yamamoto

**Affiliations:** 1Department of Microbiology, Tokai University School of Medicine, 143 Shimokasuya, Isehara 259-1193, Kanagawa, Japan; 2cmud015@mail.u-tokai.ac.jp (M.T.); tateishi.koichiro.f@tokai.ac.jp (K.T.); 3cmud009@mail.u-tokai.ac.jp (K.H.); katou.chikara.n@tokai.ac.jp (C.K.); susumu.takekoshi@nifty.ne.jp (S.T.); 2Department of Emergency and Critical Care Medicine, Tokai University School of Medicine, 143 Shimokasuya, Isehara 259-1193, Kanagawa, Japan; 3Department of Life Science Support, Research Innovation Center, University Hospitals Sector, Tokai University, 143 Shimokasuya, Isehara 259-1193, Kanagawa, Japan; n-fuku@tokai.ac.jp (N.F.); s.yusuke0220@tokai.ac.jp (Y.S.); 4Department of Rehabilitation, Osaka Kawasaki Rehabilitation University, 158 Mizuma, Kaizuka 597-0104, Osaka, Japan; konor@kawasakigakuen.ac.jp (R.K.); utsu@kawasakigakuen.ac.jp (H.U.); 5Department of Applied Chemistry and Biochemistry, National Institute of Technology, Wakayama College, 77 Noshima, Nada, Gobo 644-0023, Wakayama, Japan; natsumi-sorama@i.softbank.jp (S.N.); okuno@wakayama-nct.ac.jp (Y.O.)

**Keywords:** *Prunus mume*, SARS-CoV-2, influenza virus, Umeboshi, antiviral activity, virucidal mechanism, viral inactivation

## Abstract

Severe acute respiratory syndrome coronavirus 2 (SARS-CoV-2) and influenza virus are major respiratory pathogens associated with substantial morbidity and a risk of severe disease. However, the effectiveness of current vaccines and antiviral drugs is limited by viral mutations. Umeboshi, a traditional Japanese food prepared from pickled *Prunus mume*, is known for its health benefits; certain components of *P. mume* have exhibited antimicrobial properties. However, the efficacy of *P. mume* against SARS-CoV-2 and influenza virus remains unknown. We aimed to examine the antiviral activity of *P. mume* extracts against SARS-CoV-2 and influenza virus. Cytopathic effect (CPE) assays and reverse transcription–quantitative polymerase chain reaction (RT-qPCR) analyses with full-time treatment demonstrated that four extracts (PM2, PM3, PM4, and PM6) among eight tested inhibited the replication of both viruses. Subsequent time-of-addition assays, plaque assays, and transmission electron microscopy (TEM) confirmed that PM2 directly inactivated viral particles of both viruses by disrupting their structural integrity. Additional evaluations of virion integrity and infectivity suggested that the antiviral activity of PM2 may also involve mechanisms other than direct virion disruption. These findings suggest that *P. mume*-derived components exhibit direct antiviral activities against SARS-CoV-2 and influenza virus, supporting their potential development as antiviral agents or infection-preventive dietary products.

## 1. Introduction

Humans are continuously exposed to a wide range of pathogens, including viruses and bacteria. Among these, novel or mutated viruses have repeatedly triggered global pandemics, posing serious threats to public health. A recent example is the outbreak of severe acute respiratory syndrome coronavirus 2 (SARS-CoV-2) in late 2019, which has led to an ongoing global pandemic with hundreds of millions of confirmed cases worldwide [1,2,3,4].

In addition to SARS-CoV-2, influenza viruses represent another major group of respiratory pathogens [5,6,7,8,9,10,11]. These viruses are particularly prevalent during the winter season, causing approximately one billion infections annually. Influenza infections result in significant morbidity and mortality, particularly among high-risk populations such as infants and the elderly.

Currently, several antiviral agents are available for treating these viral infections. Approved therapeutics for SARS-CoV-2 include remdesivir and nirmatrelvir, which target viral RNA polymerase and main protease, respectively [12]. For influenza virus infections, neuraminidase inhibitors such as oseltamivir and zanamivir and cap-dependent endonuclease inhibitor baloxavir marboxil are commonly prescribed [13,14]. Vaccination also plays a crucial role in preventing severe disease outcomes [15,16,17]. However, the emergence of viral variants can compromise the efficacy of both antivirals and vaccines. Therefore, non-pharmaceutical interventions, such as handwashing, mask-wearing, and gargling, remain important for infection prevention. In this context, the identification and use of foods with antiviral properties that can be safely and routinely consumed may offer a practical and accessible approach to preventing infection and mitigating symptoms.

A variety of foods have been reported to exhibit antiviral potential against human pathogenic viruses. For example, papaya leaves, which have been consumed as food and traditional herbs, exhibit anti-coronaviral activities as well as anti-inflammatory effects [18]. Similarly, *Panax notoginseng*, used as a traditional medicine and as a functional food, contains dammarane-type triterpenoid saponins that have been shown to exert anti-inflammatory, anti-angiogenic, and anti-dengue virus activities [19].

*Prunus mume* (commonly known as Japanese apricot or Ume) is a traditional Japanese food, typically consumed in the form of Umeboshi, a salt-pickled and sun-dried preparation. It has long been recognized as a health-promoting food, with reported benefits such as relief from fatigue, stimulation of appetite, regulation of gastrointestinal function, and antimicrobial effects [20,21,22,23,24,25]. Previous studies have shown its efficacy against various pathogens, including herpes simplex virus, enterohemorrhagic *Escherichia coli* O157, and *Helicobacter pylori* [26,27,28,29,30,31,32,33,34]. However, its antiviral potential against SARS-CoV-2 and influenza virus has not yet been fully explored.

We hypothesized that *P. mume* extracts may exhibit antiviral activity against respiratory viruses, specifically SARS-CoV-2 and influenza viruses. This study aimed to identify natural compounds active against these pathogens that could serve as functional dietary products or lead compounds for antiviral drug development. In this report, we evaluated the antiviral effects of *P. mume* extracts and found that certain preparations could directly inactivate viral particles, highlighting their potential as antiviral agents or infection-preventive dietary products.

## 2. Results

### 2.1. Preparation of P. mume Extracts for Antiviral Evaluation

Salt-pickled and sun-dried *P. mume* fruits (Umeboshi) and the brine exuded during the pickling process (Umezu) were used to prepare extracts (Figure 1A). Liquid–liquid extraction yielded five extracts (PM1–PM5) from the Umeboshi and three extracts (PM6–PM8) from the Umezu (Figure 1B,C). Each extract was lyophilized and dissolved in DMSO.

### 2.2. PM2, PM3, PM4, and PM6 Inhibit Both SARS-CoV-2 and Influenza Virus Replication upon Treatment of Both Viruses and Host Cells

As an initial assessment of whether *P. mume* extracts (PM1–PM8) exhibit antiviral activity at any step of the viral life cycle, we conducted cytopathic effect (CPE) assays and reverse transcription quantitative PCR (RT-qPCR) analyses by exposing both viruses and host cells to the extracts at a concentration of 200 µg/mL throughout the entire infection cycle (Figure 2A). PM2, PM3, PM4, and PM6 significantly reduced SARS-CoV-2-induced CPE at 48 h post-inoculation and viral RNA levels at 24 h post-inoculation (Figure 2B,C). Similar results were obtained in the experiments with influenza virus, where PM2, PM3, PM4, and PM6 also suppressed CPE and reduced viral RNA levels (Figure 2D,E). These results indicate that PM2, PM3, PM4, and PM6 effectively inhibit the replication of both SARS-CoV-2 and influenza virus in vitro.

### 2.3. Time-of-Addition Assays to Identify Targets and Stages of Action of Active P. mume Extracts

Since PM2, PM3, PM4, and PM6 suppressed viral replication when both viruses and host cells were continuously exposed to *P. mume* extracts, we performed time-of-addition assays to determine the target (virus or host cell) and the stage of action (pre-entry, entry, or post-entry) of these extracts. We first examined the effects of treating virus particles with *P. mume* extracts during the pre-entry stage (Figure 3A). Pretreatment of SARS-CoV-2 with PM2 or PM3 markedly inhibited viral infection compared to the condition without pretreatment (Figure 3B,C). Similarly, pretreatment of influenza virus with PM2 and PM4 suppressed infection more effectively than the condition without pretreatment. These findings indicate that PM2 potently inhibits the pre-entry stage of both SARS-CoV-2 and influenza virus. In contrast, PM3 and PM4 selectively inhibit the pre-entry stage of SARS-CoV-2 and influenza virus infection, respectively.

Next, we conducted additional time-of-addition assays to evaluate the effects of *P. mume* extracts on host cells during the pre-entry, entry, and post-entry stages (Figure 4A). Treatment of host cells with PM2, PM3, or PM6 during the pre-entry stage significantly suppressed SARS-CoV-2 replication, whereas none of the extracts inhibited influenza virus infection (Figure 4B). When host cells and viruses were treated simultaneously during the entry stage, PM2, PM3, and PM4 significantly inhibited SARS-CoV-2 replication, while no inhibition was observed for influenza virus (Figure 4C). Post-infection treatment of host cells with PM2, PM3, or PM6 reduced replication of both SARS-CoV-2 and influenza virus (Figure 4D). All extracts tested inhibited SARS-CoV-2 replication when virions were treated during entry in combination with host-cell treatment extending from entry to post-entry (Figure 4E) or from pre-entry to post-entry (Figure 4F). Under these conditions, PM4 showed no inhibitory effect on influenza virus replication, whereas the other extracts significantly suppressed it (Figure 4E,F).

### 2.4. PM2 and PM3 Exhibit Direct Antiviral Activity Against SARS-CoV-2 Virions

The time-of-addition assays revealed that the treatment of viruses with PM2 and PM3 during the pre-entry stage was the most effective to inhibit SARS-CoV-2 infection. To further determine whether the antiviral activity was exerted directly on the viruses or mediated via the host cells, we performed plaque assays which enable assessment of the direct virucidal activity of the extracts. PM2 and PM3 significantly inhibited plaque formation of the ancestral strain, the Delta variant (B.1.617.2), and the Omicron (BA.5) variant of SARS-CoV-2 at a concentration of 200 µg/mL (Figure 5A). PM2 showed EC_50_ values of 2.06 µg/mL, 6.90 µg/mL, and 8.49 µg/mL against the ancestral strain, Delta, and Omicron, respectively. In contrast, PM3 exhibited EC_50_ values of 131.0 µg/mL, 51.60 µg/mL, and 31.26 µg/mL for the corresponding viral strains (Figure 5B). These results identify PM2 as the most potent extract with direct antiviral activity against SARS-CoV-2 among those tested.

### 2.5. PM2 and PM4 Exhibit Direct Antiviral Activity Against Influenza Virus

In addition to SARS-CoV-2, plaque assays for influenza virus were performed to evaluate direct virucidal activity of *P. mume* extracts. PM2 exhibited the strongest antiviral activity among the eight extracts against A(H1N1)pdm09, A(H3N2), and B/Yamagata viruses (Figure 6A). In contrast to SARS-CoV-2, PM4, rather than PM3, showed inhibitory effects on influenza virus plaque formation. EC_50_ values were subsequently determined for PM2 and PM4. PM2 exhibited EC_50_ values of 0.95 µg/mL, 2.60 µg/mL, and 2.17 µg/mL against A(H1N1)pdm09, A(H3N2), and B/Yamagata viruses, respectively. In comparison, PM4 exhibited EC_50_ values of 13.59 µg/mL, 53.20 µg/mL, and 48.83 µg/mL for the corresponding viruses (Figure 6B). These results indicate that PM2 possessed broad and potent virucidal activity against both SARS-CoV-2 and influenza viruses.

### 2.6. P. mume Extracts Are Non-Cytotoxic to VeroE6/TMPRSS2 and MDCK Cells up to 200 µg/mL

Given that virucidal compounds frequently exhibit cytotoxic effects, the viability of VeroE6/TMPRSS2 and MDCK cells treated with *P. mume* extracts was evaluated using an MTS assay. No significant cytotoxicity was observed for any of the eight extracts at concentrations up to 200 µg/mL (Figure 7A,B). These results indicate that the active extracts PM2, PM3, and PM4 combine potent antiviral efficacy with minimal cytotoxicity.

### 2.7. EC_50_, EC_90_, CC_50_, and Selectivity Index (SI) of Active P. mume Extracts

EC_90_ values were calculated using a four-parameter logistic model in the same manner as EC_50_ and CC_50_ values (Table 1). PM2 showed EC_90_ values of 7.23 µg/mL, 20.87 µg/mL, and 16.74 µg/mL against SARS-CoV-2 ancestral strain, Delta variant, and Omicron variant, respectively. For influenza viruses, PM2 exhibited EC_90_ values of 23.77 µg/mL, 23.71 µg/mL, and 13.05 µg/mL against A(H1N1)pdm09, A(H3N2), and B/Yamagata viruses, respectively. Based on the EC_50_ and CC_50_ values, SIs of active *P. mume* extracts were calculated (Table 1). Among them, PM2 exhibited SIs ranging from 97.09 to 23.56 for SARS-CoV-2 and from 209.4 to 76.92 for influenza viruses. These results indicate that PM2 possesses both potent antiviral activity and favorable safety profile.

### 2.8. PM2 Directly Damages SARS-CoV-2 and Influenza Virus Particles

As PM2 exhibited potent antiviral activity against both SARS-CoV-2 and influenza viruses, its direct effects on virion structure were investigated by transmission electron microscopy (TEM) and immunoelectron microscopy (IEM). In control samples, spherical virions with spike proteins or hemagglutinins were found (Figure 8A) and gold particles were consistently localized on the surfaces of morphologically intact virions, indicating preservation of their structural integrity (Figure 8B). In contrast, PM2-treated samples frequently exhibited virions with deformed or collapsed morphology (Figure 8A). Even in particles that had lost their structural integrity, gold colloids were detected in the surrounding area, suggesting they were disintegrated virions (Figure 8B). These observations indicate that PM2 disrupts the structural integrity of both SARS-CoV-2 and influenza viruses, supporting a direct virucidal mechanism.

To provide additional evidence of the damaging effect of PM2 on the viral envelope, the extent of envelope disruption in SARS-CoV-2 and influenza virus was quantitatively assessed using RNase-based virion integrity assays. In PM2-treated samples, detectable viral RNA levels were significantly reduced compared to controls, indicating that the viral envelopes were compromised, thereby exposing the viral RNA genomes to RNase A digestion (Figure 9A,D). The reduction in viral RNA levels was approximately 0.5 log_10_ units for SARS-CoV-2 and 1.5 log_10_ units for the influenza virus, suggesting greater susceptibility of the influenza virus to PM2-induced structural damage. To further examine the relationship between virion disruption and loss of infectivity, we compared the results of virion integrity assay with viral titers determined by broad-range plaque assay. This modified plaque assay was performed using not only 100-fold diluted samples but also 10-fold and undiluted samples to expand the quantification range. PM2 treatment led to a reduction in virus titers exceeding 4 log_10_ units, substantially greater than the reduction observed in the virion integrity assay (Figure 9B,C,E,F). These findings suggest that the antiviral activity of PM2 may involve additional mechanisms other than physical disruption of virions.

### 2.9. Gas Chromatograpy–Mass Spectrometry (GC-MS) Analysis of PM2 to Identify Antiviral Compounds

GC-MS analysis was performed to examine the chemical constituents of PM2. A total of 52 peaks were detected in the total ion chromatogram (Figure 10), and 44 compounds were identified by matching the generated mass spectra with entities in the NIST library (Table 2). The list of compounds mainly included fatty acids, fatty acid esters, aliphatic alcohols, steroids, and terpenoids. The seven most abundant components (>1%) were as follows: hexadecanoic acid (44.34%), 9,12-octadecadienoic acid (27.14%), methyl hexadecanoate (6.67%), methyl 9,12-octadecadienoate (5.66%), β-Sitosterol (2.86%), cycloeucalenyl acetate (1.48%), and methyl-9,12,15-octadecatrienoate (1.38%).

## 3. Discussion

Among the tested *P. mume* extracts, PM2, PM3, PM4, and PM6 exhibited antiviral activity against both SARS-CoV-2 and influenza virus, as determined by the full-time CPE/RT-qPCR assays (Figure 2). However, the plaque assays revealed a more selective antiviral profile: PM2 and PM3 inhibited SARS-CoV-2 infectivity, whereas PM4 and PM6 showed little or no such effect (Figure 5). Regarding influenza virus, PM2 and PM4 exhibited strong antiviral activity in plaque assays, while PM3 and PM6 were ineffective or only slightly effective (Figure 6). These findings indicate that the antiviral activities of *P. mume* extracts vary not only among the individual extracts but also between the two virus species.

The observed discrepancies between the full-time CPE/RT-qPCR assays and the plaque assays can be attributed to the distinct mechanisms assessed by each method. The full-time CPE/RT-qPCR analyses evaluate the overall antiviral effect, reflecting potential interactions of *P. mume* extracts with viral particles and host cells, as the extracts remain present throughout the infection process. In contrast, the plaque assay specifically assesses the direct impact on viral infectivity, as the virions are incubated with *P. mume* extracts only before infection. This distinction suggests that some extracts, such as PM6, may exert their antiviral effects through host-dependent mechanisms rather than by directly inactivating viral particles.

Time-of-addition assays revealed that PM2 exerts its antiviral activity through multiple mechanisms. PM2 significantly inhibited both SARS-CoV-2 and influenza virus infection when virions were treated during the pre-entry stage and when host cells were treated during the post-entry stage (Figure 3 and Figure 4). These findings suggest that PM2 acts on both virus particles and host cells. Notably, because the most pronounced inhibition was observed when virions were treated during the pre-entry stage, the primary antiviral mechanism of PM2 is most likely the direct inactivation of viral infectivity. Since both PM2 and neutralizing antibodies act on virions before infection and inhibit viral entry, PM2 may serve as a potential substitute for neutralizing antibodies.

The ability of PM2 to reduce the infectivity of both SARS-CoV-2 and influenza virus in the plaque assays suggests a direct interaction with a viral component common to both viruses. Given that both possess lipid bilayer envelopes, it is plausible that PM2 exerts its antiviral effects by disrupting this shared structural feature. This mechanism is supported by the TEM and IEM analyses, which revealed that PM2 directly damages the viral envelopes of both viruses (Figure 8). Despite this membrane-disrupting activity, PM2 demonstrated minimal cytotoxicity toward host cells (Figure 7). Mammalian cells possess membrane repair mechanisms that enable recovery from transient damage, whereas viruses lack such capacity and are irreversibly inactivated by membrane disruption [35,36,37,38,39].

PM3 selectively reduced SARS-CoV-2 infectivity in the plaque assay while exhibiting minimal or no effect against the influenza virus (Figure 5 and Figure 6). PM3 inhibited SARS-CoV-2 infection in the time-of-addition assays when virions were exposed during the pre-entry stage and when host cells were treated at the pre-entry, entry, or post-entry stages (Figure 3 and Figure 4). In contrast, PM3 suppressed influenza virus infection only when host cells were treated during the post-entry stage. These results indicate that PM3 acts on both viruses and host cells in SARS-CoV-2 replication, but only on host cells in influenza virus replication.

PM4 inhibited influenza virus infection in the plaque assay but showed no activity against SARS-CoV-2 (Figure 5 and Figure 6). In time-of-addition assays, PM4 suppressed SARS-CoV-2 replication only when host cells were treated during the entry stage, whereas it blocked influenza virus infection only when virions were exposed during the pre-entry stage (Figure 3 and Figure 4). These findings suggest that PM4 interferes with SARS-CoV-2 entry through host cell-related factors, while inhibiting influenza virus infection via direct virion inactivation.

PM6 was inactive in plaque assays but suppressed replication of both viruses in the full-time CPE/RT-qPCR assays, indicating that its antiviral effect depends primarily on host-mediated pathways rather than direct virion inactivation. Time-of-addition assays further showed that treatment of host cells during the post-entry stage inhibited replication of both SARS-CoV-2 and influenza virus. These host-related antiviral effects may involve activation of antiviral signaling pathways such as the interferon response, although the precise mechanisms remain to be elucidated.

PM1, the source extract from which PM2 to PM5 were derived, is expected to contain the antiviral components present in PM2, PM3, and PM4. However, PM1 itself exhibited no detectable antiviral activity, likely due to the low concentration of these active components. As shown in Figure 1B, the weights of PM2, PM3, and PM4 corresponded to only 1.2%, 1.6%, and 6.5% of the total weight of PM1, respectively.

In the virion integrity assays, PM2 reduced viral RNA levels by approximately 0.5 log_10_ units for SARS-CoV-2 and 1.5 log_10_ units for influenza virus, indicating that PM2 induces envelope damage sufficient to expose viral RNA to RNase A digestion (Figure 9A,D). However, broad-range plaque assays demonstrated a more substantial reduction in viral infectivity, exceeding 4 log_10_ units for both viruses (Figure 9B,C,E,F). This discrepancy suggests that, in addition to disrupting the viral envelope, PM2 may impair other essential viral components, such as surface glycoproteins, thereby further reducing infectivity. EC_50_ values obtained from plaque assays reveal that the influenza virus is more susceptible to PM2 than SARS-CoV-2 (Figure 5 and Figure 6), which may be attributed to structural differences, particularly in their envelope glycoproteins that influence the interaction and efficacy of PM2.

Previous studies have demonstrated that *P. mume* contains a variety of organic acids, including citric and malic acids, as well as a wide spectrum of phenolic compounds with both hydrophilic and hydrophobic characteristics [40,41,42,43,44,45,46,47,48]. Earlier research has focused on the antimicrobial properties of Umezu or phenolic fractions derived from the skin or seed. For instance, Ikeda et al. reported that phenolic compounds isolated from Umezu exhibited antiviral activity against the influenza virus at a concentration of 4000 µg/mL [34]. In contrast, the present study employed stepwise solvent extraction from both Umeboshi and Umezu using hydrophobic solvents such as hexane, dichloromethane, and ethyl acetate. PM2, obtained from the hexane extract of Umeboshi, showed the most potent antiviral activity, with EC_50_ values between 0.95 and 2.60 µg/mL against influenza virus strains. These findings suggest that PM2 contains hydrophobic antiviral components and that Umeboshi may offer a higher concentration of such compounds compared to Umezu.

In this study, GC-MS analysis was conducted to identify potential antiviral compounds present in PM2 (Figure 10 and Table 2). The results indicated that PM2 primarily composed of fatty acids, fatty acid esters, aliphatic alcohols, steroids, and terpenoids. Among these, 9,12-octadecadienoic acid (linoleic acid) was one of the major constituents, accounting for 27.4% of the total content. Linoleic acid has been reported to inactivate enveloped viruses by disrupting virion membranes [49,50,51], suggesting a key role in the membrane-disrupting activity of PM2 against viral particles. Moreover, linoleic acid has been shown to bind to the SARS-CoV-2 spike protein and inhibit viral entry by stabilizing the spike in a locked conformation that is incompatible with ACE2 binding [52]. These findings imply that linoleic acid could also be involved in the suppression of SARS-CoV-2 infection by PM2 through mechanisms other than direct membrane disruption. Regarding the steroidal components, β-sitosterol was the most abundant, comprising 2.86% of PM2. Shokry et al. reported that β-sitosterol exhibits potent antiviral activity against influenza A virus by directly interacting with the viral particles [53]. Furthermore, molecular docking studies suggested that β-sitosterol binds to hemagglutinin protein of the influenza virus [53]. These observations point to a potential role of β-sitosterol in the inhibition of influenza virus infection by PM2, likely through pathways independent of membrane disruption. Taken together, our findings along with previous reports suggest that both linoleic acid and β-sitosterol may contribute, at least in part, to the antiviral activity observed in PM2.

This study provides the first evidence that the *P. mume* extract PM2 exhibits antiviral activity against multiple strains of SARS-CoV-2 and influenza virus. While previous reports have demonstrated antimicrobial properties of *P. mume* extracts against herpes simplex virus, enterohemorrhagic *Escherichia coli* O157, and *Helicobacter pylori*, the present findings expand this antimicrobial spectrum to include pathogenic respiratory viruses. This highlights the potential of *P. mume* as a source of broad-spectrum antiviral agents.

This study has several limitations. The antiviral efficacy, pharmacokinetics, and safety of PM2 have not yet been validated in vivo. Additionally, the specific active compounds responsible for the observed antiviral effects remain unidentified, and detailed molecular interaction studies are required to elucidate the underlying mechanisms.

To address these limitations, future research should focus on isolating and characterizing the active components within PM2 and elucidating their antiviral mechanisms at the molecular level. In addition, in vivo studies using appropriate animal models, such as a mouse model for influenza virus, a hamster model for SARS-CoV-2, and a human ACE2-transgenic mouse model for SARS-CoV-2, will be essential to evaluate the pharmacokinetics, safety profile, and therapeutic potential of these compounds. Such studies may pave the way for developing novel antiviral agents or functional dietary products derived from *P. mume* with broad-spectrum antiviral activity.

## 4. Materials and Methods

### 4.1. Cells and Viruses

VeroE6/TMPRSS2 cells and Madin–Darby Canine Kidney (MDCK) cells were obtained from the Japanese Collection of Research Bioresources (JCRB; Osaka, Japan) and the American Type Culture Collection (ATCC; Rockville, MD, USA), respectively. Both cell lines were cultured in Dulbecco’s Modified Eagle’s Medium (DMEM; Thermo Fisher Scientific, Waltham, MA, USA) supplemented with 1 g/L D-glucose, 4 mM L-glutamine, 10% fetal bovine serum (FBS), and 1% penicillin/streptomycin (all from Thermo Fisher Scientific). For virus stock preparation, the following SARS-CoV-2 strains were used: WK521 (ancestral strain), TY11-927 (Delta variant, lineage B.1.617.2), and TY41-702 (Omicron variant, sublineage BA.5). These strains were propagated in VeroE6/TMPRSS2 cells cultured in Minimum Essential Medium (MEM; Thermo Fisher Scientific) supplemented with 2% FBS and 1% penicillin/streptomycin. For influenza virus propagation, the following strains were used: A/California/7/2009 (A(H1N1)pdm09 subtype), X31 (A(H3N2) subtype), and B/Florida/4/2006 (B/Yamagata lineage). These viruses were amplified in MDCK cells cultured in OptiPRO SFM (Thermo Fisher Scientific) supplemented with 4 mM L-glutamine and 2 µg/mL acetylated trypsin (Merck, Rahway, NJ, USA). All cell cultures were incubated at 37 °C in a humidified atmosphere containing 5% CO_2_ throughout the experiments.

### 4.2. Extraction of Fractions from P. mume Fruit (PM1–PM5)

Salt-pickled and sun-dried *P. mume* (Umeboshi, 4.8 kg) was soaked in 5 L of methanol at 25 °C for 24 h. The mixture was filtered to obtain the methanol extract. This extraction process was repeated twice (three times in total), and the combined methanol extracts were concentrated under reduced pressure to yield the crude methanol extract (PM1).

The crude methanol extract was suspended in 2 L of water and subjected to liquid–liquid extraction with 500 mL of hexane. This procedure was performed three times. The combined hexane layers were collected and concentrated under reduced pressure to obtain the hexane-soluble fraction (PM2).

The remaining aqueous phase was sequentially extracted with 500 mL each of dichloromethane and ethyl acetate using the same procedure. The dichloromethane-soluble (PM3) and ethyl acetate-soluble (PM4) fractions were individually collected and concentrated under reduced pressure. The final aqueous phase was also concentrated to yield the water-soluble fraction (PM5).

### 4.3. Extraction of Fractions from P. mume Pickling Brine (PM6–PM8)

Pickling brine derived from *P. mume* (Umezu) was desalted, concentrated, and suspended in water. The mixture was centrifuged at 3000 rpm for 5 min to separate the supernatant (soluble fraction) from the precipitate (insoluble fraction). The soluble fraction was subjected to liquid–liquid extraction with 500 mL of hexane, repeated three times, and the combined hexane extracts were concentrated under reduced pressure.

The remaining aqueous phase was sequentially extracted with 500 mL of dichloromethane and ethyl acetate following the same procedure. The dichloromethane-soluble (PM6) and ethyl acetate-soluble (PM7) fractions were collected and concentrated under reduced pressure. The final aqueous phase was concentrated to yield the water-soluble fraction (PM8).

Among the fractions obtained from Umezu, all except the hexane-soluble fraction were used in subsequent experiments.

### 4.4. Cytopathic Effect (CPE) Assay with Full-Time Treatment of Viruses and Host Cells with P. mume Extracts

*P. mume* extracts were dissolved in dimethyl sulfoxide (DMSO; FUJIFILM Wako Pure Chemical Corporation, Osaka, Japan) at a 40 mg/mL concentration and subsequently diluted 100-fold in MEM. Equal volumes of the diluted extract and virus solution (SARS-CoV-2 or influenza virus at 2000 PFU/mL in MEM) were mixed to obtain final concentrations of 200 µg/mL for the extract and 1000 PFU/mL for the virus. The mixtures were incubated at 25 °C for 1 h to allow interaction.

VeroE6/TMPRSS2 cells (for SARS-CoV-2) or MDCK cells (for influenza virus) were seeded in 96-well plates and washed three times with MEM prior to treatment. The cells were pretreated with 200 µg/mL *P. mume* extract in MEM at 37 °C for 30 min. After removing the extract solution, the virus-extract mixtures were added to the cells at a multiplicity of infection (MOI) of 0.01 and incubated at 37 °C for 1 h. The inoculum was then removed and replaced with a virus replication medium containing 200 µg/mL of *P. mume* extract. For VeroE6/TMPRSS2 cells, the replication medium consisted of MEM supplemented with 2% FBS. For MDCK cells, a 1:1 mixture of MEM and OptiPRO SFM supplemented with 4 mM L-glutamine and 2 µg/mL acetylated trypsin was used.

Cells were incubated at 37 °C for 48 h, and CPEs were observed using a BZ-X810 microscope (Keyence Corporation, Osaka, Japan).

### 4.5. RNA Extraction

The viruses and the cells were treated with *P. mume* extracts as described in CPE assay or time-of-addition assay, and 50 µL of the culture supernatant was collected at 12 h or 24 h post-inoculation. The samples were mixed with 0.9 mL TRI Reagent (Molecular Research Center, Cincinnati, OH, USA) containing carrier RNA (total RNA from uninfected VeroE6/TMPRSS2 cells). Viral RNA and carrier RNA were extracted according to the manufacturer’s instructions.

### 4.6. Reverse Transcription Quantitative PCR (RT-qPCR)

Viral RNA was quantified using the TaqMan Fast Virus 1-step Real-Time RT-PCR assay kit (Thermo Fisher Scientific) and the QuantStudio 3 Real-Time PCR system (Thermo Fisher Scientific), as previously described [54,55,56], with modifications to the reaction setup to adapt it to our system. The primer and probe sequences, along with the reaction conditions, were as follows:

For SARS-CoV-2:Forward primer: NIID_2019-nCOV_N_F2, 5′-AAATTTTGGGGACCAGGAAC-3′Reverse primer: NIID_2019-nCOV_N_R2v3, 5′-TGGCACCTGTGTAGGTCAAC-3′Probe: NIID_2019-nCOV_N_P2, 5′-FAM-ATGTCGCGCATTGGCATGGA-BHQ-3′Thermal cycling: 55 °C for 10 min, 95 °C for 3 min, followed by 40 cycles at 95 °C for 15 s and 58 °C for 30 s.

For influenza virus and 18S ribosomal RNA:
Influenza forward primer: FluV-F, 5′-CACCTGATATTGTGGATTACTGATCG-3′Influenza reverse primer: FluV-R, 5′-CACTCTGCTGTTCCTGTTGATATTC-3′Influenza probe: FluV-P, 5′-FAM-CCTCATGGACTCAGGCACTCCTTCCG-TAMRA-3′18S forward primer: 18S-F, 5′-GTAACCCGTTGAACCCCATT-3′18S reverse primer: 18S-R, 5′-CCATCCAATCGGTAGTAGCG-3′18S probe: 18S-P, 5′-FAM-TGCGTTGATTAAGTCCCTGCCCTTTGTA-TAMRA-3′Thermal cycling: 50 °C for 5 min, 95 °C for 20 s, followed by 40 cycles at 95 °C for 1 s and 60 °C for 20 s.

### 4.7. Time-of-Addition Assay

To evaluate the effect of pretreating virions with *P. mume* extracts, these extracts were mixed with SARS-CoV-2 and influenza virus at a final extract concentration of 200 µg/mL and a final virus concentration of 2000 PFU/mL. The mixtures were incubated at 25 °C for 1 h and then VeroE6/TMPRSS2 and MDCK cells were exposed to the mixtures at an MOI of 0.01 for 1 h at 37 °C. To remove viruses and *P. mume* extracts, the cells were washed with serum-free MEM three times and cultured with the replication medium.

To assess the effect of pretreating host cells with the extracts, the cells were incubated with the extracts (200 µg/mL) for 30 min, washed with serum-free MEM three times, and infected with SARS-CoV-2 or influenza virus (MOI 0.01).

To evaluate the effect of simultaneous treatment of viruses and host cells during the entry stage, the virus-extract mixtures were inoculated immediately after preparation. The cells were exposed to the mixtures (MOI 0.01), washed with serum-free MEM three times, and cultured with the replication medium.

To investigate the effect of treating host cells during the post-entry stage, the cells were infected with SARS-CoV-2 or influenza virus (MOI 0.01), washed with serum-free MEM three times, and cultured with replication medium containing 200 µg/mL of *P. mume* extracts.

The supernatants were collected at 12 h or 24 h post-inoculation for RT-qPCR analysis.

### 4.8. Plaque Assay

VeroE6/TMPRSS2 (for SARS-CoV-2) or MDCK cells (for influenza virus) were seeded in 6-well plates. *P. mume* extracts were serially diluted and mixed with SARS-CoV-2 or influenza virus to yield final extract concentrations ranging from 200 to 0.39 µg/mL and a final virus concentration of 1 × 10^5^ PFU/mL. The mixtures were incubated at 25 °C for 1 h to allow reaction. For standard plaque assays, the virus-extract mixtures were diluted 100-fold in MEM. For broad-range plaque assays, undiluted, 10-fold diluted, and 100-fold diluted mixtures were prepared. Subsequently, 200 µL of each virus-extract mixture was added to each well. The cells were incubated at 37 °C for 1 h to allow virus adsorption.

Following infection, the inoculum was removed and replaced with 2 mL of plaque assay overlay medium. For SARS-CoV-2, the overlay medium consisted of MEM supplemented with 1.2% colloidal cellulose (Merck, Rahway, NJ, USA) and 2% FBS. For the influenza virus, the overlay medium consisted of MEM supplemented with 0.6% agar (Bacto agar; Becton, Dickinson and Company, Franklin Lakes, NJ, USA), 0.2% bovine serum albumin (Merck), and 10 µg/mL acetylated trypsin.

The cells were incubated at 37 °C for 3 days. After incubation, the overlay medium was removed, and the cells were washed twice with PBS. Fixation was performed using 4% paraformaldehyde phosphate-buffered solution (PFA; Nacalai Tesque, Kyoto, Japan) at 4 °C for 30 min. The cells were then stained with crystal violet (FUJIFILM Wako Pure Chemical Corporation, Osaka, Japan), and plaques were counted.

### 4.9. MTS Assay

VeroE6/TMPRSS2 and MDCK cells were treated with serial dilutions of *P. mume* extracts ranging from 200 to 0.78 µg/mL at 37 °C for 24 h. After treatment, 20 µL of MTS reagent (CellTiter 96 AQueous One Solution Cell Proliferation Assay; Promega, Madison, WI, USA) was added to each well of a 96-well plate. The cells were then incubated at 37 °C for 4 h, and absorbance was measured at 490 nm using a SpectraMax i3 microplate reader (Molecular Devices, San Jose, CA, USA).

### 4.10. Transmission Electron Microscopy (TEM) of Virions Treated with PM2

Virus stocks were diluted in PBS and treated with the *P. mume* extract PM2 or DMSO in PBS at 25 °C for 1 h. The final concentrations during treatment were 100 µg/mL for PM2, 1 × 10^7^ PFU/mL for SARS-CoV-2, and 1 × 10^8^ PFU/mL for influenza virus. After incubation, an equal volume of 8% PFA was added to inactivate the viruses. The samples were negatively stained with 1% phosphotungstic acid (PTA) or 1% uranyl acetate (UA), and analyzed by TEM using a JEM-1400 microscope (JEOL, Tokyo, Japan).

### 4.11. Immunoelectron Microscopy (IEM) of Virions Treated with PM2

PM2-treated or DMSO-treated virus samples were fixed with PFA and incubated with a monoclonal antibody against the SARS-CoV-2 S protein (GTX632604; Gene Tex, Irvine, CA, USA) or a monoclonal antibody against the influenza virus HA protein (mAb3; [57]). The samples were then incubated with a 10 nm gold colloid-conjugated secondary antibody (EMGMHL10; BBI Solutions, Crumlin, UK). After antibody labeling, negative staining was performed using 1% PTA or 1% UA. The localization of viral antigens was visualized by TEM using a JEM-1400 microscope.

### 4.12. Virion Integrity Assay

To evaluate virion integrity, PM2 solution or control solution (PBS with or without DMSO) was mixed with diluted SARS-CoV-2 or influenza virus to achieve final concentrations of 100 µg/mL for PM2 and 1.25 × 10^5^ PFU/mL for each virus. The mixtures were incubated at 25 °C for 1 h. Subsequently, RNase A was added to a final concentration of 1 µg/mL, and the samples were further incubated at 37 °C for 30 min to degrade exposed viral RNA. Viral RNA was then extracted using the Direct-zol RNA miniprep kit (Zymo Research, Irvine, CA, USA) according to the manufacturer’s instructions. The amount of intact viral RNA, protected from RNase A digestion by an undamaged viral envelope, was quantified by RT-qPCR.

### 4.13. Statistical Analysis and Calculation of 50% Effective Concentrations (EC_50_)

All values are expressed as the mean ± standard deviation (SD). Statistical differences were evaluated using Dunnett’s multiple comparison tests and paired Student’s *t*-tests, and *p*-values < 0.05 were considered statistically significant. The EC_50_ values were calculated using GraphPad Prism 9 software (GraphPad Software, Boston, MA, USA) using a four-parameter logistic curve.

### 4.14. Gas Chromatography–Mass Spectrometry (GC-MS) Analysis

GC-MS analysis was performed using an Agilent 7890A gas chromatograph (Agilent Technologies, Santa Clara, CA, USA) coupled with a JMS-Q1050 mass spectrometer (JEOL).

Separation was achieved using a capillary column HP-5MS (30 m × 0.25 mm id; film thickness 0.25 μm; Agilent technologies). The oven temperature was programmed to increase from 100 °C to 300 °C at a rate of 4 °C/min and held at 300 °C for 18 min. Helium was used as the carrier gas at a constant flow rate of 1.2 mL/min. The injector, transfer line, and ion source temperatures were set at 250 °C. The ionization energy was 70 eV. The mass spectra were recorded over an m/z range of 40–550. A 1 µL sample was injected in splitless mode. Compound identification was carried out by comparison with spectra from the NIST library.

## Figures and Tables

**Figure 1 ijms-26-08487-f001:**
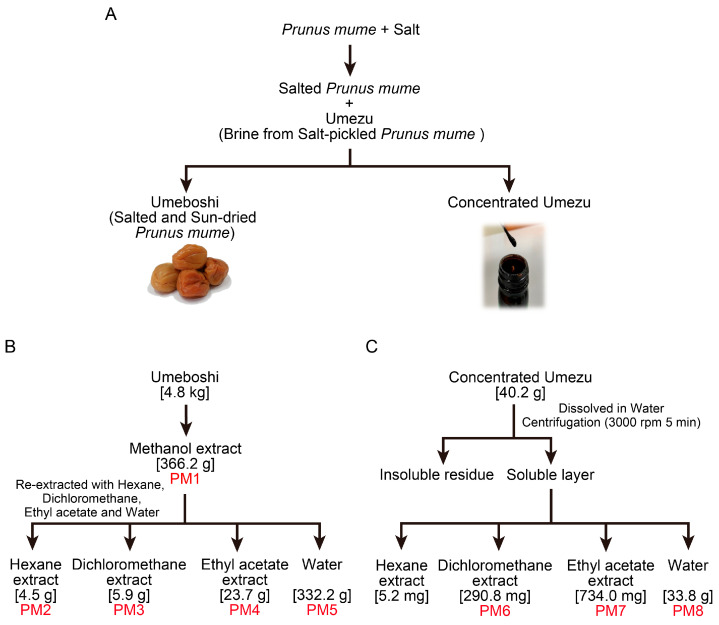
Preparation of *Prunus mume* extracts PM1–PM8 by liquid–liquid partitioning. (**A**) Umeboshi (salt-pickled and sun-dried *P. mume*) and Umezu (the brine generated during the pickling process) were prepared from *P. mume* fruits. (**B**) PM1–PM5 were extracted from Umeboshi using methanol, hexane, dichloromethane, ethyl acetate, and water. (**C**) PM6–PM8 were extracted from the soluble fraction of Umezu with dichloromethane, ethyl acetate, and water. Each extract was lyophilized and dissolved in DMSO.

**Figure 2 ijms-26-08487-f002:**
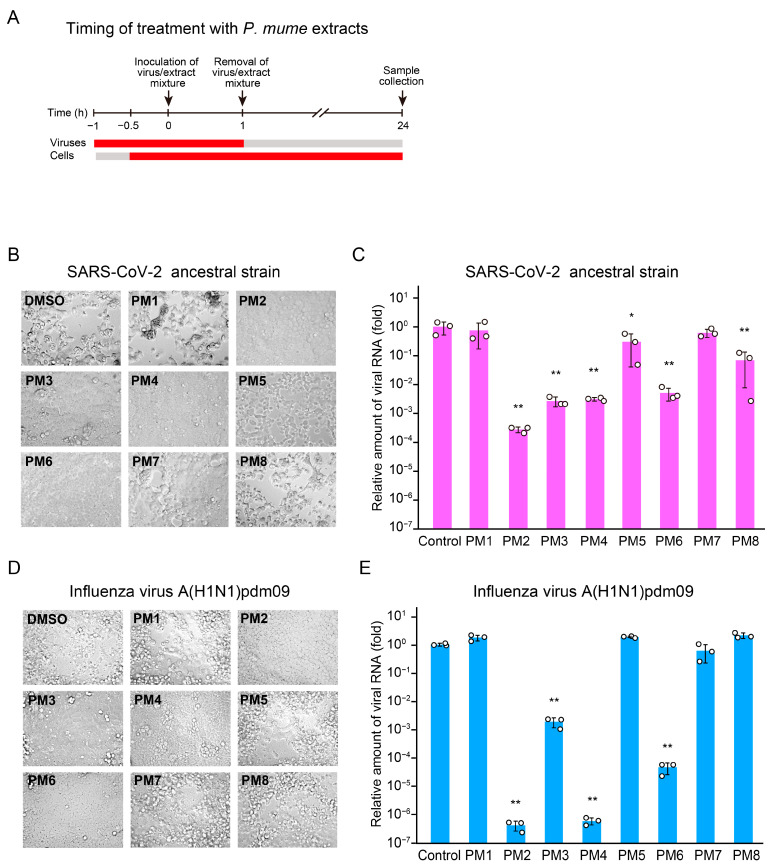
Antiviral effects of *P. mume* extracts against SARS-CoV-2 and influenza virus following full-time treatment of both viruses and host cells evaluated by CPE assays and RT-qPCR analyses. (**A**) Schematic representation of the timing of *P. mume* extract treatment of viruses and cells. (**B**,**D**) Representative CPE in VeroE6/TMPRSS2 cells exposed to SARS-CoV-2/*P. mume* extract mixtures (**B**) and MDCK cells exposed to influenza virus/*P. mume* extract mixtures (**D**), both at an MOI of 0.01. Viruses and cells were treated with *P. mume* extracts (PM1–PM8, 200 µg/mL), and images were captured using microscopy at 48 h post-inoculation. (**C**,**E**) Viral RNA levels in the culture supernatants treated under the same condition as in (**B**,**D**), quantified by RT-qPCR at 24 h post-inoculation. Data are presented as the mean ± standard deviation (SD) of three independent experiments. Statistical significance was assessed using Dunnett’s multiple comparison test, with the untreated control group serving as the reference. * *p* < 0.05, ** *p* < 0.01.

**Figure 3 ijms-26-08487-f003:**
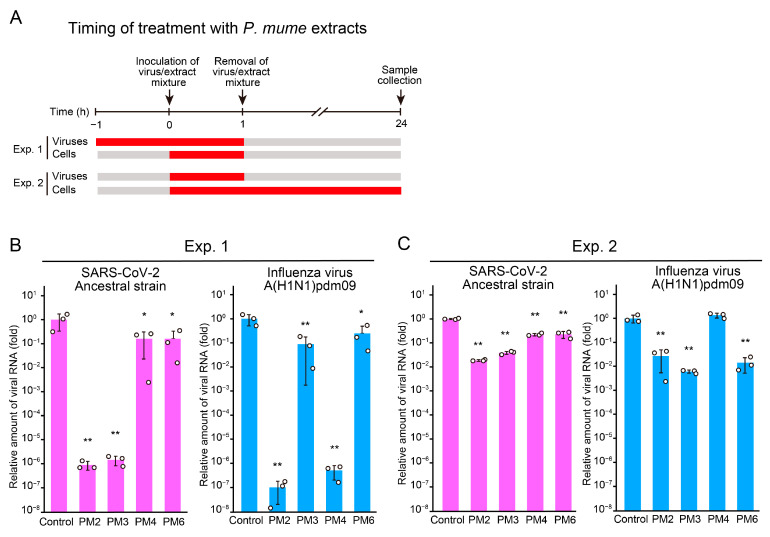
Antiviral effects of *P. mume* extracts against SARS-CoV-2 and influenza virus evaluated by time-of-addition assay for virus particles. (**A**) Schematic representation of the timing of *P. mume* extract treatment of viruses and cells. (**B**,**C**) Viral RNA levels in the culture supernatants at 24 h post-inoculation were quantified by RT-qPCR analysis for SARS-CoV-2 and influenza virus. Viruses were treated with active *P. mume* extracts (PM2, PM3, PM4, and PM6) during both the pre-entry and entry stages (**B**), or during the entry stage only (**C**). Data are presented as the mean ± SD of three independent experiments. Statistical significance was assessed using Dunnett’s multiple comparison test, with the untreated control group serving as the reference. * *p* < 0.05, ** *p* < 0.01.

**Figure 4 ijms-26-08487-f004:**
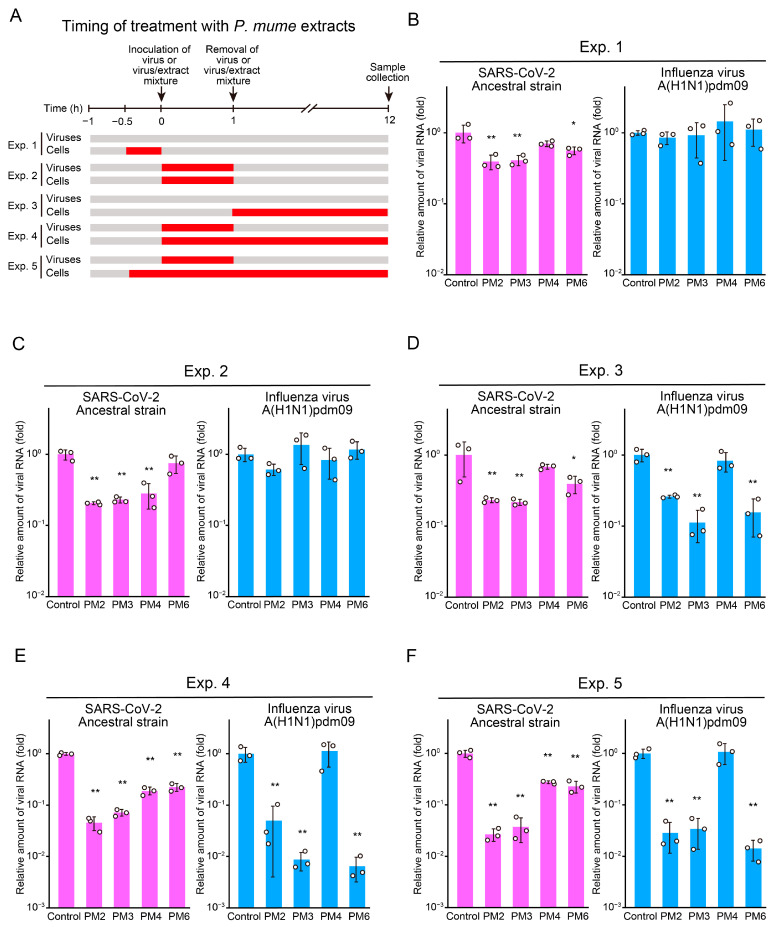
Antiviral effects of *P. mume* extracts against SARS-CoV-2 and influenza virus evaluated by time-of-addition assay for host cells. (**A**) Schematic representation of the timing of *P. mume* extract treatment of viruses and cells. (**B**–**F**) Viral RNA levels in the culture supernatants at 12 h post-inoculation were quantified by RT-qPCR analysis for SARS-CoV-2 and influenza virus. Host cells were treated with *P. mume* extracts (PM2, PM3, PM4, and PM6) during the pre-entry stage (**B**), the entry stage (**C**), the post-entry stage (**D**), from entry to post-entry (**E**), or from pre-entry to post-entry (**F**). Data are presented as the mean ± SD of three independent experiments. Statistical significance was assessed using Dunnett’s multiple comparison test, with the untreated control group serving as the reference. * *p* < 0.05, ** *p* < 0.01.

**Figure 5 ijms-26-08487-f005:**
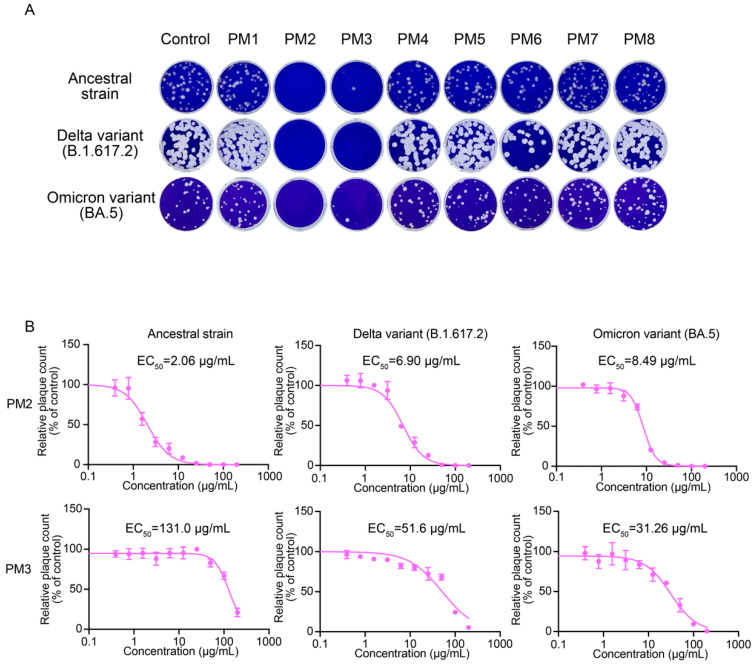
Antiviral effects of *P. mume* extracts against SARS-CoV-2 evaluated by plaque assay. (**A**) Representative plaque assay images showing SARS-CoV-2 treated with PM1–PM8 at a 200 µg/mL concentration. (**B**) EC_50_ values of PM2 and PM3 were determined by plaque assays. Plaques were counted, and relative plaque counts were calculated as percentages of the untreated virus control. Data are presented as the mean ± SD from three independent experiments. EC_50_ values were determined by a four-parameter logistic model.

**Figure 6 ijms-26-08487-f006:**
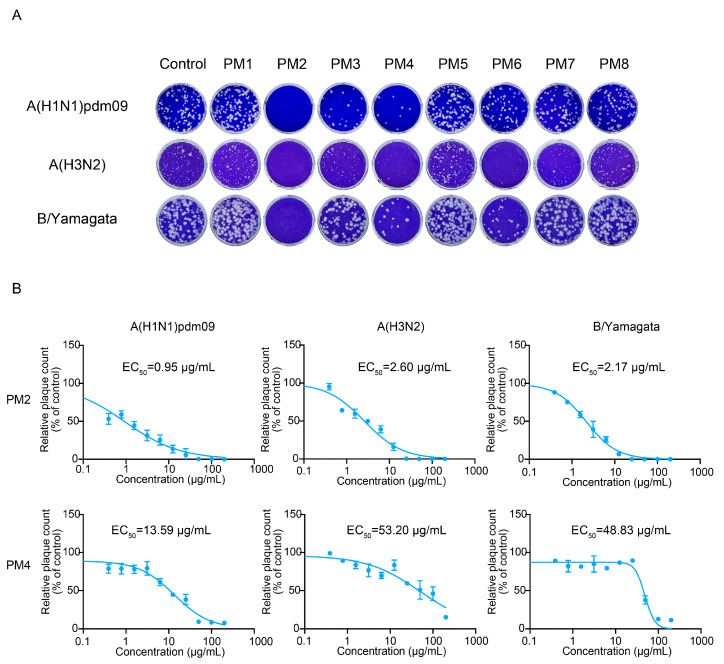
Inhibitory effects of *P. mume* extracts against influenza viruses assessed by plaque assay. (**A**) Representative plaque assay results for influenza viruses treated with PM1–PM8 at 200 µg/mL. (**B**) EC_50_ values of PM2 and PM4 were determined by plaque assays. Plaque counts were normalized to control values. Data are presented as the mean ± SD from three independent experiments. EC_50_ values were calculated using a four-parameter logistic model.

**Figure 7 ijms-26-08487-f007:**
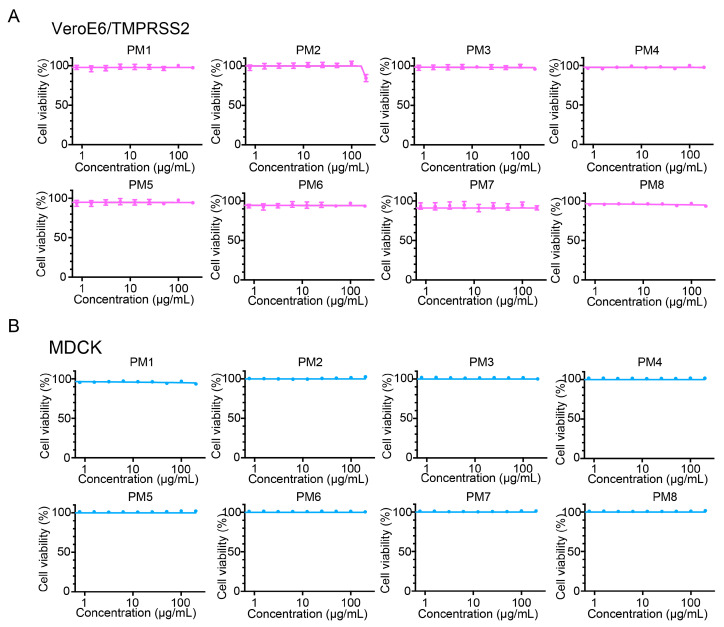
Cytotoxicity of *P. mume* extracts in VeroE6/TMPRSS2 and MDCK cells assessed by MTS assay. (**A**) Viability of VeroE6/TMPRSS2 cells and (**B**) MDCK cells following treatment with *P. mume* extracts at concentrations ranging from 0 to 200 µg/mL. Relative cell viability was calculated as a percentage of the untreated control. Data are presented as the mean ± SD from three independent experiments.

**Figure 8 ijms-26-08487-f008:**
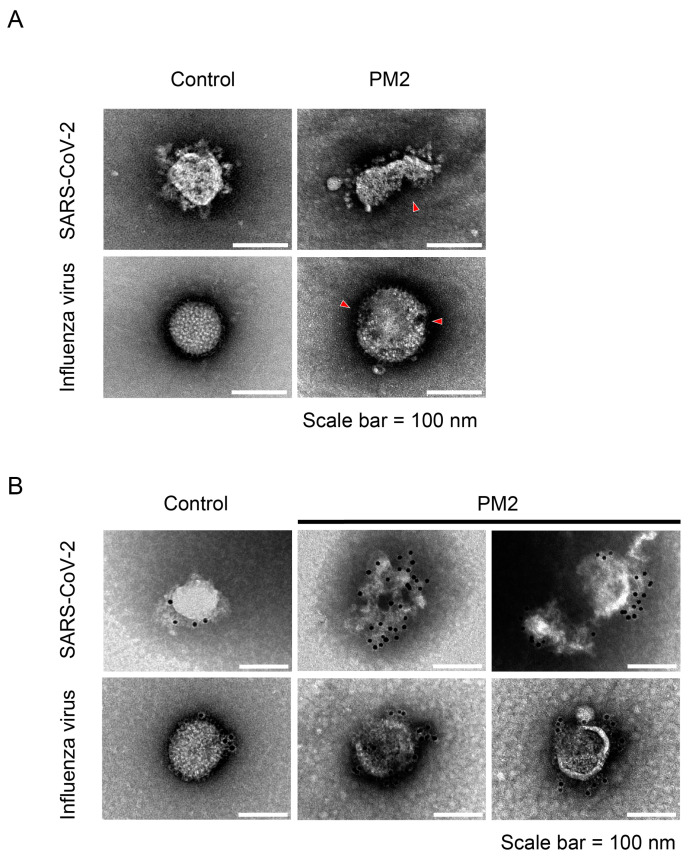
Effects of PM2 on SARS-CoV-2 and influenza virus particles as evaluated by transmission electron microscopy (TEM) and immunoelectron microscopy (IEM). (**A**) Structural damages to SARS-CoV-2 (ancestral strain) and influenza virus (A(H1N1)pdm09) virions were assessed by TEM. Red arrows indicate areas of apparent morphological disruption compared to the control. (**B**) Virion integrity was further analyzed by IEM using virus-specific primary antibodies and gold-labeled secondary antibodies. The black dots represent gold colloids bound to viral spike or hemagglutinin proteins.

**Figure 9 ijms-26-08487-f009:**
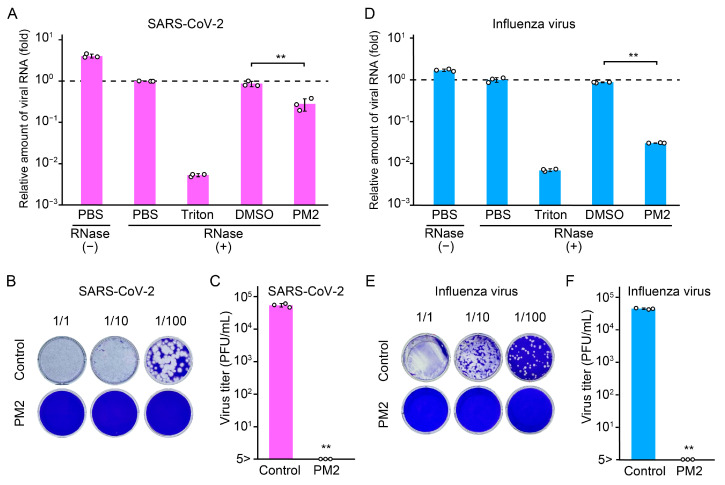
Effects of PM2 on viral envelope structure and infectivity evaluated by virion integrity and broad-range plaque assay. (**A**,**D**) Assessment of PM2-induced damage to SARS-CoV-2 (**A**) and influenza virus (**D**) using a virion integrity assay. Treatment with Triton plus RNase A served as the positive control (indicating complete virion disruption), while phosphate-buffered saline (PBS) plus RNase A served as the negative control, reflecting RNA levels within intact virions protected from RNase A degradation. (**B**,**E**) Representative broad-range plaque assay images of SARS-CoV-2 (**B**) and influenza virus (**E**) treated with either PM2 or DMSO in PBS. (**C**,**F**) Virus titers of SARS-CoV-2 (**C**) and influenza virus (**F**) were determined by broad-range plaque assays following treatment with PM2 or DMSO in PBS. Data are presented as the mean ± SD from three independent experiments. Statistical significance was determined by Student’s *t*-test, comparing PM2-treated samples with the DMSO control. ** *p* < 0.01.

**Figure 10 ijms-26-08487-f010:**
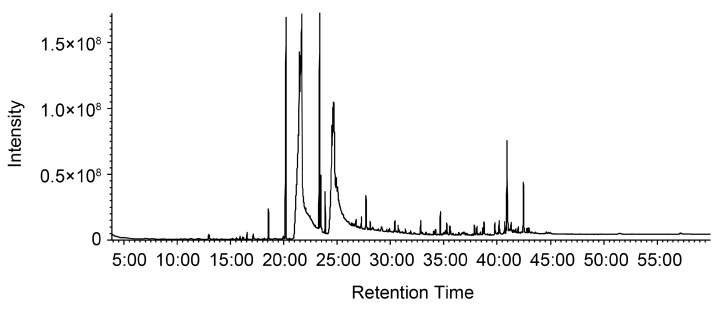
Total ion chromatogram of PM2, the hexane extract from *P. mume*, obtained by GC-MS analysis.

**Table 1 ijms-26-08487-t001:** EC_50_, EC_90_, CC_50_, and Selectivity Index of active *P. mume* extracts against SARS-CoV-2s and influenza viruses.

Extract	Virus	EC_50_ (µg/mL)	EC_90_ (µg/mL)	CC_50_ (µg/mL)	Selectivity Index
PM2	SARS-CoV-2 Ancestral strain	2.06	7.23	>200	>97.09
PM2	SARS-CoV-2 Delta variant (B.1.617.2)	6.90	20.87	>200	>28.99
PM2	SARS-CoV-2 Omicron variant (BA.5)	8.49	16.74	>200	>23.56
PM3	SARS-CoV-2 Ancestral strain	131.0	>200	>200	>1.53
PM3	SARS-CoV-2 Delta variant (B.1.617.2)	51.60	>200	>200	>3.88
PM3	SARS-CoV-2 Omicron variant (BA.5)	31.26	131.3	>200	>6.40
PM2	Influenza virus A(H1N1)pdm09	0.95	23.77	>200	>209.4
PM2	Influenza virus A(H3N2)	2.60	23.71	>200	>76.92
PM2	Influenza virus B/Yamagata	2.17	13.05	>200	>92.17
PM4	Influenza virus A(H1N1)pdm09	13.59	110.2	>200	>14.72
PM4	Influenza virus A(H3N2)	53.20	>200	>200	>3.76
PM4	Influenza virus B/Yamagata	48.83	85.38	>200	>4.10

**Table 2 ijms-26-08487-t002:** Chemical components of PM2 identified by GC-MS analysis.

No.	RT	Compounds	MF	MW	Area
1	12:57	Dodecanoic acid	C_12_H_24_O_2_	200.32	0.25
2	15:32	Hexadecylene oxide	C_16_H_32_O	240.42	0.03
3	15:52	Hexadecanal	C_16_H_32_O	240.42	0.083
4	16:06	Methyl tetradecanoate	C_15_H_30_O_2_	242.40	0.06
5	16:31	Octadecyl vinyl ether	C_20_H_40_O	296.53	0.15
6	17:06	Tetradecanoic acid	C_14_H_28_O_2_	228.37	0.20
7	18:31	Hexahydrofarnesyl acetone	C_18_H_36_O	268.48	0.62
8	20:00	Methyl (7E)-7-hexadecenoate	C_17_H_32_O_2_	268.43	0.07
9	20:10	Methyl hexadecanoate	C_17_H_34_O_2_	270.45	6.67
10	21:39	Hexadecanoic acid	C_16_H_32_O_2_	256.42	44.34
11	23:19	Methyl 9,12-octadecadienoate	C_19_H_34_O_2_	294.47	5.66
12	23:24	Methyl-9,12,15-octadecatrienoate	C_19_H_32_O_2_	292.46	1.38
13	23:32	Methyl octadecanoate	C_19_H_38_O_2_	298.50	0.06
14	24:39	9,12-Octadecadienoic acid	C_18_H_32_O_2_	280.45	27.14
15	26:30	8,11,14-Eicosatrienoic Acid	C_20_H_34_O_2_	306.48	0.03
16	26:39	Methyl 11-(3-pentyl-2-oxiranyl)undecanoate	C_19_H_36_O_3_	312.49	0.06
17	26:44	Eicosane	C_20_H_42_	282.55	0.15
18	27:15	Methyl eicosanoate	C_21_H_42_O_2_	326.56	0.30
19	27:41	4,8,12,16-Tetramethylheptadecan-4-olide	C_21_H_40_O_2_	324.54	0.83
20	28:04	Butyl hexadecanoate	C_20_H_40_O_2_	312.53	0.18
21	28:19	Ethyl docosanoate	C_24_H_48_O_2_	368.64	0.06
22	30:17	Farnesyl acetate	C_17_H_28_O_2_	264.40	0.05
23	30:22	Methyl 5-(2-undecylcyclopropyl)pentanoate	C_20_H_38_O_2_	310.51	0.28
24	30:41	Methyl 9-(2-[(2-butylcyclopropyl)methyl]cyclopropyl)nonanoate	C_21_H_38_O_2_	322.53	0.18
25	31:52	2-Tetradecen-1-ol	C_14_H_28_O	212.37	0.07
26	32:48	2-Hexyldecanol	C_16_H_34_O	242.44	0.36
27	33:18	Methyl heptacosanoate	C_28_H_56_O_2_	424.74	0.06
28	34:01	9-Octadecenamide	C_18_H_35_NO	281.48	0.07
29	34:39	Supraene	C_30_H_50_	410.72	0.50
30	35:08	Unknown			0.10
31	35:15	Unknown			0.33
32	35:32	Eicosane	C_20_H_42_	282.55	0.19
33	35:35	Octadecanol	C_18_H_38_O	270.49	0.11
34	35:49	Unknown			0.04
35	36:23	Unknown (sterols)			0.05
36	36:50	Heptacosane	C_27_H_56_	380.73	0.04
37	36:56	Heptatriacotanol	C_37_H_76_O	537.00	0.02
38	37:02	Unknown			0.03
39	37:09	Geranylgeraniol	C_20_H_34_O	290.48	0.03
40	37:50	Cholesta-4,6-dien-3-ol	C_27_H_44_O	384.64	0.25
41	38:05	Stigmastan-3,5-diene	C_29_H_48_	396.69	0.31
42	38:25	α-Tocopherol	C_29_H_50_O_2_	430.71	0.07
43	38:44	Unknown			0.33
44	39:46	Campesterol	C_28_H_48_O	400.68	0.38
45	40:10	Stigmasterol	C_29_H_48_O	412.69	0.42
46	40:55	β-Sitosterol	C_29_H_50_O	414.71	2.86
47	41:05	24-Propylidenecholesterol	C_30_H_50_O	426.72	0.04
48	41:17	Cycloeucalenyl acetate	C_32_H_52_O_2_	468.75	0.25
49	41:43	Cycloartenyl acetate	C_32_H_52_O_2_	468.75	0.07
50	41:57	Stigmasta-3,5-dien-7-one	C_29_H_46_O	410.67	0.14
51	42:26	Cycloeucalenyl acetate	C_32_H_52_O_2_	468.75	1.48
52	42:47	Cholesta-4,6-dien-3-one	C_27_H_42_O	382.62	0.12
		Total			100.00

RT: retention time; MF: molecular formula; MW: molecular weight.

## Data Availability

The datasets analyzed during the current study are available from the corresponding author upon reasonable request.

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
