# Peer review of "Prunus mume Extract Inhibits SARS-CoV-2 and Influenza Virus Infection In Vitro by Directly Targeting Viral Particles"

_ijms, 2025, doi:10.3390/ijms26178487_

Round 1
Reviewer 1 Report
Comments and Suggestions for Authors
The work turns out to be very interesting: the study of the antiviral activity of P. Mume extracts, thus natural molecules that we can eat is very timely and useful for the future.
However, I would have some notes to make:
- the introduction: it seems to me to be a bit too general and unrelated to the data reported later. I would like it to be more about the drugs currently in use, the antiviral activity of natural molecules such as P. Mume extracts can be, and less about the characteristics of the viruses used that we all know by now. Because little leaks out from this introduction about the purpose of this article: is the aim to identify the natural molecule with antiviral activity so as to prepare extracts that can be taken as a supplement? Or to find molecules in general with antiviral activity against these two viruses?
- In Figure 2: The 2B graph and the 2D graph could be made with the same scale? So that we can understand well the difference of action on the two viruses.
Also, in this figure, the statistic used in the caption is not marked. Was it used throughout the t-test? Why, for example, in Figure 2B, do PM1 and PM2 have the same significance despite being so different?
Finally, the caption is not well understood, because there is a double description of point A and point C. Isn't rt-qPCR described in section 2.6 and figure 7? Is this an error or is it the photo of the samples also used for RT-qPCR?
Perhaps try using a multiple comparison statistics method such as Bonferroni or Dunnett (slightly more correct than a t-test, as it avoids the correction with FDR that should be done with t-test, but which I do not see written in the materials and methods). - For section 2.3 and 2.4, can we have a table with well described EC50s, EC90s, CC50s, and SIs of these extracts?
- How come the treatment type was changed from CPE to plaque assay?
If the goal was to find a molecule that acts in the pre-infection or during infection, thus in the stages of virus attachment or entry, why do a full-time CPE?
It would also be interesting to see the curves of PM4 and PM6 in section 2.3 and of PM2, PM4 and PM6 in section 2.4 to actually see the difference between the two treatments. - You observed that PM2 extract acts directly on the virion by changing its morphology, also speculating that it may interact with membrane glycoproteins. Why not do an attachment assay and an EMSA (using, for example, recombinant spike for SARS-CoV-2 and HA for IAV to be incubated with PM2 for different timings or with different concentrations of PM2) to actually validate this hypothesis?
Author Response
Comment 1:
the introduction: it seems to me to be a bit too general and unrelated to the data reported later. I would like it to be more about the drugs currently in use, the antiviral activity of natural molecules such as P. Mume extracts can be, and less about the characteristics of the viruses used that we all know by now. Because little leaks out from this introduction about the purpose of this article: is the aim to identify the natural molecule with antiviral activity so as to prepare extracts that can be taken as a supplement? Or to find molecules in general with antiviral activity against these two viruses?
Response 1:
Thank you for your valuable comments. We have revised the introduction of our manuscript based on your advice. We have reduced the general description of the viruses (lines 44-54), added the information of the antiviral drugs and the natural products with antiviral activity (lines 57-59, 66-71), and described clearly the purpose of this study (lines 81-86).
Comment 2:
(1) In Figure 2: The 2B graph and the 2D graph could be made with the same scale? So that we can understand well the difference of action on the two viruses.
(2) Also, in this figure, the statistic used in the caption is not marked. Was it used throughout the t-test? Why, for example, in Figure 2B, do PM1 and PM2 have the same significance despite being so different?
Perhaps try using a multiple comparison statistics method such as Bonferroni or Dunnett (slightly more correct than a t-test, as it avoids the correction with FDR that should be done with t-test, but which I do not see written in the materials and methods).
(3) Finally, the caption is not well understood, because there is a double description of point A and point C. Isn't rt-qPCR described in section 2.6 and figure 7? Is this an error or is it the photo of the samples also used for RT-qPCR?
Response 2:
(1) Thank you for your constructive comment. We have revised Figure 2B to align its scale with Figure 2D, as you suggested. During this revision, we also identified and corrected an error in the original preparation of Figure 2B. The corrected Figure 2B has now been included.
(2) Following your suggestion, we have conducted the Dunnett's multiple comparison test instead of the Student's t-test used prior to revision. The symbols indicating the statistical significance and the caption of Figure 2 have been updated accordingly.
(3) To address your concern, we have revised the caption of Figure 2 in our manuscript.
Comment 3:
For section 2.3 and 2.4, can we have a table with well described EC50s, EC90s, CC50s, and SIs of these extracts?
Response 3:
Thank you for your helpful comment to improve our manuscript. Following your advice, we added a new table (now designated as Table 1) describing EC50s, EC90s, CC50s, and SIs of active extracts.
Comment 4:
How come the treatment type was changed from CPE to plaque assay?
If the goal was to find a molecule that acts in the pre-infection or during infection, thus in the stages of virus attachment or entry, why do a full-time CPE?
It would also be interesting to see the curves of PM4 and PM6 in section 2.3 and of PM2, PM4 and PM6 in section 2.4 to actually see the difference between the two treatments.
Response 4:
We sincerely appreciate your valuable feedback. In reviewing the manuscript in light of your suggestion, we realized that the reason for initially performing the CPE assay and subsequently conducting the plaque assay was not clearly explained. We have revised the manuscript to clarify this point (lines 104-108, 130-133, 152-156).
Comment 5:
You observed that PM2 extract acts directly on the virion by changing its morphology, also speculating that it may interact with membrane glycoproteins. Why not do an attachment assay and an EMSA (using, for example, recombinant spike for SARS-CoV-2 and HA for IAV to be incubated with PM2 for different timings or with different concentrations of PM2) to actually validate this hypothesis?
Response 5:
Thank you very much for your thoughtful and constructive suggestion. We fully agree that attachment assays and EMSA using recombinant spike (SARS-CoV-2) or hemagglutinin (IAV) proteins would provide direct evidence to support our hypothesis regarding the interaction between PM2 and viral glycoproteins. However, due to the unavailability of recombinant spike and HA proteins in our laboratory, we were unable to perform these specific experiments. As an alternative approach, we conducted time-of-addition assays to investigate the effect of PM2 on the early stages of viral infection, including pre-entry and entry steps (Figure 3). In addition, we have expanded our discussion about linoleic acid and β-sitosterol, which have been identified in PM2 by our GC-MS analysis and have been reported to interact with the spike and HA proteins, respectively (lines 371-377, 379-384). These findings support the possibility that certain PM2 components, such as linoleic acid and β-sitosterol, may directly interact with viral glycoproteins.
Reviewer 2 Report
Comments and Suggestions for Authors
Prunus mume extract inhibits SARS-CoV-2 and influenza virus infection in vitro by directly targeting viral particles
The authors of “Prunus mume extract inhibits SARS-CoV-2 and influenza virus infection in vitro by directly targeting viral particles” present interesting findings showing that Prunus mume extract PM2 exhibits in vitro antiviral activity against multiple SARS-CoV-2 variants and influenza virus strains, possibly via direct virucidal action. The study combines CPE assays, plaque assays, TEM, IEM, and GC-MS profiling, which together provide a broad view of the potential antiviral mechanism. However, authors should conduct a substantial revision to address the following comments, specially regarding the PM2s’ mechanism of action in its antiviral activity to strengthen their findings.
Major comments
- Though authors listed the lack of in vivo data as a limitation, the manuscript would benefit from adding one of the widely available IAV mouse models to validate the claims of potential therapeutic use of the PM2 extract.
- As indicated in lines 381-384, the authors pre-treated the cells with each extract before virus and extract mixture treatment. This treatment diverges from the authors' finding of PM2-mediated viral structural damage. Among the many active compounds in PM2 extract, some may promote the IFN signaling pathway to enhance the host's innate immune response and suppress virus replication. Therefore, authors should check the IFN pathway activation by PM2 to further validate their claims about the PM2s' mechanism of action.
- Among the active compounds discovered from the HPLC, the authors discussed Linoleic acid and β-Sitosterol in lines 302-309 in the discussion section. However, neither Linoleic acid nor β-Sitosterol is known to induce viral particle damage. The references given by the authors also show that Linoleic acid is involved in spike and ACE2 interaction inhibition, and β-Sitosterol inhibits host cell entry of IAV by interacting with HA, NA, and M2. Furthermore, in lines 202-203, the authors suggest the antiviral activity of PM2 may involve additional mechanisms other than physical disruption of virions. Authors should address these questions about PM2s’ mechanism of action using a time of addition assay to find the exact step of the virus life cycle that was targeted by PM2, which further strengthens these results.
- Authors should discuss the reason for PM1 not having any effect on CPE or viral RNA amount in Figures 2-4.
- Authors should conduct a CC50 assay with higher doses of PM2 and calculate the selectivity index (SI), which is critical to evaluate the therapeutic potential.
- While DMSO is used as the solvent for PM2 extract, its independent effect on viral particles was not assessed in the TEM or IEM.
Minor comments
- Figures 3 and 4: EC50 values indicated in the text are different from values in the figures
Author Response
Major comments
Comment 1:
Though authors listed the lack of in vivo data as a limitation, the manuscript would benefit from adding one of the widely available IAV mouse models to validate the claims of potential therapeutic use of the PM2 extract.
Response 1:
Thank you very much for your valuable comment. We totally agree with your suggestion. However, due to the limitation of time for revision of the manuscript, we were unable to add the data of IAV mouse model experiments. Instead, we have added the discussion about animal experiments to validate the efficacy of PM2 (lines 399-402).
Comment 2:
As indicated in lines 381-384, the authors pre-treated the cells with each extract before virus and extract mixture treatment. This treatment diverges from the authors' finding of PM2-mediated viral structural damage. Among the many active compounds in PM2 extract, some may promote the IFN signaling pathway to enhance the host's innate immune response and suppress virus replication. Therefore, authors should check the IFN pathway activation by PM2 to further validate their claims about the PM2s' mechanism of action.
Response 2:
Thank you very much for pointing this out. As an initial assessment of whether P. mume extracts exhibit antiviral activity at any step of the viral life cycle, we performed full-time CPE assays and RT-qPCR analyses in which both viruses and host cells were exposed to the extracts throughout the entire infection cycle. To further narrow down the target of PM2, we then conducted plaque assays to evaluate whether PM2 exerts direct effects on virus particles. The plaque assays demonstrated that PM2 exhibited strong antiviral activity even in the absence of host cell treatment, suggesting that PM2 can inhibit virus infection independently of IFN signaling. In addition, we performed a time-of-addition assay without pre-treatment of host cells with the extracts. The results also support the notion that PM2 can inhibit viral replication without requiring prior induction of IFN. We have included an explanation for initially performing the CPE assay/RT-qPCR analysis (lines 104-108) and a discussion about the IFN (lines 336-338) in the revised manuscript.
Comment 3:
Among the active compounds discovered from the HPLC, the authors discussed Linoleic acid and β-Sitosterol in lines 302-309 in the discussion section. However, neither Linoleic acid nor β-Sitosterol is known to induce viral particle damage. The references given by the authors also show that Linoleic acid is involved in spike and ACE2 interaction inhibition, and β-Sitosterol inhibits host cell entry of IAV by interacting with HA, NA, and M2. Furthermore, in lines 202-203, the authors suggest the antiviral activity of PM2 may involve additional mechanisms other than physical disruption of virions. Authors should address these questions about PM2s’ mechanism of action using a time of addition assay to find the exact step of the virus life cycle that was targeted by PM2, which further strengthens these results.
Response 3:
We sincerely appreciate your comments which have strengthen our work. We have found at least three papers that reported the membrane-damaging activity of linoleic acid against enveloped viruses. We have included these papers as references in the revised manuscript (lines 371-373, references 49-51). Furthermore, we performed a time-of-addition assay in response to your suggestion (Figure 3). The sections of Results, Discussion, and Materials and Methods have been revised accordingly.
Comment 4:
Authors should discuss the reason for PM1 not having any effect on CPE or viral RNA amount in Figures 2-4.
Response 4:
Thank you for your insightful comment. We have added the discussion about the activity of PM1 (lines 339-343).
Comment 5:
Authors should conduct a CC50 assay with higher doses of PM2 and calculate the selectivity index (SI), which is critical to evaluate the therapeutic potential.
Response 5:
Due to the limitation of reagents, it is difficult to perform CC50 assay with higher doses of P. mume extracts. Within the range of 0-200 µg/mL, we calculated SIs and put a table to show them in accordance with your suggestion (Table 1).
Comment 6:
While DMSO is used as the solvent for PM2 extract, its independent effect on viral particles was not assessed in the TEM or IEM.
Response 6:
Thank you for insightful comment. We have assessed the effect of DMSO in TEM and IEM, and replaced the previous images of PBS-treated samples with those of DMSO-treated samples in the revised manuscript (Figure 7).
Minor comments
Comment 1:
Figures 3 and 4: EC50 values indicated in the text are different from values in the figures.
Response 1:
Thank you for pointing out the errors we made. We corrected the EC50 values in the text (lines 159-160, 178-180).
Round 2
Reviewer 1 Report
Comments and Suggestions for Authors
I greatly appreciate the changes made and the improvements to the article.
The introduction is more complete, Figure 2 is clearer, but above all, the table with all the values greatly simplifies reading and gives an immediate idea of the action of these extracts.
However, there is one point that is not at all clear to me; I cannot understand how the experiments were carried out.
From the first draft, I understood that:
- CPE was a full-time experiment: pre-treatment of cells with the extract, infection with virus + extract, post-infection with extract at a fixed concentration (paragraph 2.2 and figure 2)
- plaque assay: with pre-treatment only
Rereading it now, I understand that:
- for CPE: the virus was incubated with the extract for one hour and then the cells were infected, while the cells were pre-treated for 30 minutes and then infected with the mixture. Is that correct?
Why were the cells also pre-treated? Did you happen to do a simple antiviral activity curve? Because in my opinion, that's the data that matters. An experiment done this way does not give me real antiviral activity, because you exposed the virus to the molecule before exposing it to the cells, so it does not mimic what can happen in an organism in the slightest.
- For the plaque assay, was a sort of virucidal test performed? So, was the virus incubated with the molecule and then the mixture diluted to the point that the extract no longer had antiviral activity, and with different dilution of that you infected the cells? Was that how the experiment was conducted? Because at this point it would be more correct to call it a “virucidal test” rather than a plaque assay.
- Finally, the TOA was an excellent idea to test the action of the extract during the attachment or entry phase, but here too, I did not fully understand how it was done.
Normally, the TOA is done in the following way: the cells are exposed to the extract only during pre-treatment, only during infection, or only during post-infection, in order to understand which phase the molecule acts on. Here, I do not seem to see this experimental design, but I see again a virus incubated with the extract for one hour and then placed on the cells for the second hour. Therefore, it is not possible to understand whether the effect is virucidal, whether the extract acts during the attachment or entry phase.
In conclusion, if my understanding is correct. To complete the article, an antiviral curve is needed, obtained by exposing the cells to an untreated virus, i.e. a perfectly active virus, in order to understand the real antiviral activity of these compounds. Furthermore, in my opinion, a complete TOA is necessary, including a graph showing the activity observed during pre-treatment, co-treatment or post-treatment, because this seems to me to be the simplest way to identify the phase in which the molecule acts, thus avoiding EMSA or attachment assays.
Author Response
Comment 1:
I greatly appreciate the changes made and the improvements to the article. The introduction is more complete, Figure 2 is clearer, but above all, the table with all the values greatly simplifies reading and gives an immediate idea of the action of these extracts. However, there is one point that is not at all clear to me; I cannot understand how the experiments were carried out.
From the first draft, I understood that:
- CPE was a full-time experiment: pre-treatment of cells with the extract, infection with virus + extract, post-infection with extract at a fixed concentration (paragraph 2.2 and figure 2)
- plaque assay: with pre-treatment only
Rereading it now, I understand that:
- for CPE: the virus was incubated with the extract for one hour and then the cells were infected, while the cells were pre-treated for 30 minutes and then infected with the mixture. Is that correct?
Response 1:
Thank you very much for your valuable comments. Regarding full-time CPE assay, your understanding is correct. We modified the manuscript to improve its clarity (lines 489-490).
Comment 2:
Why were the cells also pre-treated? Did you happen to do a simple antiviral activity curve? Because in my opinion, that's the data that matters. An experiment done this way does not give me real antiviral activity, because you exposed the virus to the molecule before exposing it to the cells, so it does not mimic what can happen in an organism in the slightest.
Response 2:
Thank you very much for your helpful comments. Our understanding is that an antiviral effect is classified into at least three effects: (1) prophylactic effect on viruses, (2) prophylactic effect on cells, and (3) therapeutic effect on cells.
An example of (1) is a neutralizing antibody, which binds to virus particles before infection and inhibits viral entry.
Regarding (2), an anti-CD4 antibody for HIV infection is an example, which binds to CD4 on cells before infection and blocks viral infection.
As for (3), neuraminidase inhibitor like oseltamivir inhibits the step of viral release, which means that this drug targets the late post-entry step of virus life cycle and is still active when it is added after virus infection.
In our study, full-time CPE/RT-qPCR assays were performed to initially assess whether P.mume extracts have any of prophylactic effects on viruses, prophylactic effects on cells, or therapeutic effects on cells. Therefore, we included pre-treatment of cells with the extracts in the full-time assays.
We believe that pre-treatment of viruses reflects a situation in which a prophylactically administered antiviral agent, such as a neutralizing antibody, binds to the virus prior to cellular entry, thereby preventing viral replication.
Since we are particularly interested in prophylactic effects on viruses (that is, virucidal effects), we conducted plaque assays as well as time-of-addition assays with treatment of virions during the pre-entry step. However, in response to your comment, we realized the lack of experiments using untreated viruses. Thus, we have performed time-of-addition assays without pre-treatment of viruses (Figure 3C, Figure 4B, 4C, 4D, 4E, and 4F) and revised the manuscript accordingly.
Comment 3:
- For the plaque assay, was a sort of virucidal test performed? So, was the virus incubated with the molecule and then the mixture diluted to the point that the extract no longer had antiviral activity, and with different dilution of that you infected the cells? Was that how the experiment was conducted? Because at this point it would be more correct to call it a “virucidal test” rather than a plaque assay.
Response 3:
We fully agree with your comments. Yes, we performed plaque assays to test the virucidal effect (or prophylactic effect on viruses) of P. mume extracts. The Materials and Methods section has been modified to improve its readability (lines 563-570).
Comment 4:
- Finally, the TOA was an excellent idea to test the action of the extract during the attachment or entry phase, but here too, I did not fully understand how it was done.
Normally, the TOA is done in the following way: the cells are exposed to the extract only during pre-treatment, only during infection, or only during post-infection, in order to understand which phase the molecule acts on. Here, I do not seem to see this experimental design, but I see again a virus incubated with the extract for one hour and then placed on the cells for the second hour. Therefore, it is not possible to understand whether the effect is virucidal, whether the extract acts during the attachment or entry phase.
Response 4:
We sincerely appreciate your constructive comments. Since we have an interest in the virucidal activity (prophylactic effect on viruses) of the extracts, we performed plaque assays to evaluate direct effects to virions and time-of-addition assays to assess the effects on virions during the pre-entry stage. However, in the process of revising this manuscript based on your comments, we noticed the absence of studies without pre-treatment of viruses. Therefore, we have performed additional time-of-addition assays with cells exposed to the extracts only during the pre-entry, entry, or post-entry stage as you suggested.
Comment 5:
In conclusion, if my understanding is correct. To complete the article, an antiviral curve is needed, obtained by exposing the cells to an untreated virus, i.e. a perfectly active virus, in order to understand the real antiviral activity of these compounds. Furthermore, in my opinion, a complete TOA is necessary, including a graph showing the activity observed during pre-treatment, co-treatment or post-treatment, because this seems to me to be the simplest way to identify the phase in which the molecule acts, thus avoiding EMSA or attachment assays.
Response 5:
Thank you very much for providing such helpful comments. Based on your feedback, we have included (1) antiviral assays without pre-treatment of viruses and (2) complete time-of-addition assays showing the activity of the extracts during the pre-treatment, co-treatment, or post-treatment (Figure 3C, Figure 4B, 4C, 4D, 4E, and 4F). The target (virus or host cell) and the stage (pre-entry, entry, or post-entry) of the active P. mume extracts have been analyzed comprehensively with the results of full-time CPE/RT-qPCR assays, time-of-addition assays, and plaque assays. We believe that inclusion of the results of time-of-addition assays has significantly improved the quality of our research. We thank you again for your constructive and valuable suggestion.
Reviewer 2 Report
Comments and Suggestions for Authors
The authors of the manuscript “Prunus mume extract inhibits SARS-CoV-2 and influenza virus infection in vitro by directly targeting viral particles” have adequately addressed all of the questions and concerns that were raised. After reviewing their revisions and responses, I find that the current version of the manuscript meets the necessary standards and is acceptable for publication
Author Response
Comment 1:
The authors of the manuscript “Prunus mume extract inhibits SARS-CoV-2 and influenza virus infection in vitro by directly targeting viral particles” have adequately addressed all of the questions and concerns that were raised. After reviewing their revisions and responses, I find that the current version of the manuscript meets the necessary standards and is acceptable for publication.
Response 1:
Your comments were very helpful to improve the quality of our manuscript. Thank you very much for your vital contribution.
Round 3
Reviewer 1 Report
Comments and Suggestions for Authors
Dear authors, I really appreciate the effort you have made to address my comments.
I think that the text is much clearer now and the purpose of your research is much more evident. All these experiments clearly show that the compound acts mainly on the virus, even though some effects are seen after treatment on the cells (as explained in the conclusions).
It was not clear in the previous version that the main purpose was to use this compound as a substitute for “neutralising antibodies, which bind to virus particles before infection and inhibit viral entry”, as explained in the response to my comments. In my opinion, a similar sentence could be included in the conclusions to emphasise the possible use that you wanted to investigate, so as to eliminate any doubt.
Congratulations on the work you have done.
Author Response
Comment 1:
Dear authors, I really appreciate the effort you have made to address my comments.
I think that the text is much clearer now and the purpose of your research is much more evident. All these experiments clearly show that the compound acts mainly on the virus, even though some effects are seen after treatment on the cells (as explained in the conclusions).
It was not clear in the previous version that the main purpose was to use this compound as a substitute for “neutralising antibodies, which bind to virus particles before infection and inhibit viral entry”, as explained in the response to my comments. In my opinion, a similar sentence could be included in the conclusions to emphasise the possible use that you wanted to investigate, so as to eliminate any doubt.
Congratulations on the work you have done.
Response 1:
Thank you very much for your valuable comments. Following your advice, we have revised Discussion section (lines 344-346). We sincerely appreciate your constructive inputs and important contributions, which have greatly helped us improve the quality of our manuscript through these three rounds of revision.